Microgeographic variation in locomotor traits among lizards in a human-built environment

Donihue Colin colin.donihue@yale.edu
School of Forestry and Environmental Studies, Yale University , New Haven, CT , USA
Roper James
Electronic publication date: 2016 Mar 10
Publication date: 2016
Volume: 4
Electronic Location ID: e1776
Received 2015 Sep 30; Accepted 2016 Feb 18
Copyright: ©2016 Donihue
Copyright year: 2016
Copyright holder: Donihue
License: This is an open access article distributed under the terms of the Creative Commons Attribution License, which permits unrestricted use, distribution, reproduction and adaptation in any medium and for any purpose provided that it is properly attributed. For attribution, the original author(s), title, publication source (PeerJ) and either DOI or URL of the article must be cited.
License URL: https://creativecommons.org/licenses/by/4.0/

Keywords: Locomotion, Morphometrics, Context-dependence, Lizard, Podarcis erhardii, Local adaptation, Sprint speed

Funding: National Geographic Waitt Foundation Yale Institute for Biospheric Studies Funding provided by National Geographic Waitt Foundation and the Yale Institute for Biospheric Studies. The funders had no role in study design, data collection and analysis, decision to publish, or preparation of the manuscript.

==============================
Microgeographic variation in fitness-relevant traits may be more common than previously appreciated. The fitness of many vertebrates is directly related to their locomotor capacity, a whole-organism trait integrating behavior, morphology, and physiology. Because locomotion is inextricably related to context, I hypothesized that it might vary with habitat structure in a wide-ranging lizard, Podarcis erhardii, found in the Greek Cyclade Islands. I compared lizard populations living on human-built rock walls, a novel habitat with complex vertical structure, with nearby lizard populations that are naive to human-built infrastructure and live in flat, loose-substrate habitat. I tested for differences in morphology, behavior, and performance. Lizards from built sites were larger and had significantly (and relatively) longer forelimbs and hindlimbs. The differences in hindlimb morphology were especially pronounced for distal components—the foot and longest toe. These morphologies facilitated a significant behavioral shift in jumping propensity across a rocky experimental substrate. I found no difference in maximum velocity between these populations; however, females originating from wall sites potentially accelerated faster over the rocky experimental substrate. The variation between these closely neighboring populations suggests that the lizards inhabiting walls have experienced a suite of trait changes enabling them to take advantage of the novel habitat structure created by humans.

Introduction

Animal locomotion integrates a suite of morphological, behavioral, and physiological attributes and impacts an individual’s fitness (Irschick & Garland, 2001; Calsbeek & Irschick, 2007; Irschick et al., 2008). Furthermore, locomotor behavior and performance is of necessity closely tied to an individual’s immediate ecological setting (Losos, 1990; Toro, Herrel & Irschick, 2004; Kohlsdorf & Navas, 2007). While traits associated with locomotion are often considered typological for a species, emerging evidence suggests that microgeographic variability in ecological context can result in considerably more intraspecific variation in fitness-relevant traits than previously appreciated (Richardson et al., 2014).

Other studies have demonstrated that the substrate and structure of a habitat are consistently related to a lizard species’ behavior, morphology, and performance (Vanhooydonck & Van Damme, 2003; Calsbeek & Irschick, 2007; Kohlsdorf & Navas, 2007; Losos, 2011). Lizards in more complex habitats tend to more often jump from branch-to-branch or rock-to-rock (Kohlsdorf & Navas, 2007; Harrison, Revell & Losos, 2015). Additionally, jumping performance in lizards is often associated with longer hind limbs, particularly in the distal segments between the ankle and the tip of the longest toe (Moermond, 1979; Losos, 1990; Toro, Herrel & Irschick, 2004).

Laboratory tests of lizard locomotion typically employ a single experimental substrate. Moreover, the types of substrates used may (e.g., sand) or may not (e.g., cork or sandpaper) reflect naturally occurring substrates that have given rise to different adaptations for locomotion. Comparing lizard locomotion across multiple substrates is increasingly the focus of new studies (Tulli, Abdala & Cruz, 2012; Vanhooydonck et al., 2015), but these studies have yet to investigate performance of conspecifics living in different habitats and test predictions of associated morphological and behavioral differences according to those habitats.

Humans are ecosystem engineers, creating novel habitat structure across landscapes and exerting strong adaptive pressure on the organisms in those landscapes (Jones, Lawton & Shachak, 1994; Donihue & Lambert, 2014). In the Greek islands, stone walls and terraces crisscross the landscape, and the eponymous Aegean Wall Lizard, Podarcis erhardii, can readily be found throughout (Valakos et al., 2008). However, P. erhardii, can also commonly be found dashing between bushes in nearby wall-less habitats with sand or loose-soil substrates. Podarcis erhardii living on stone walls experience a more structurally complex habitat than their conspecifics in non-wall habitats (Fig. 1). Based on other research showing that lizard traits change to accommodate new demands for locomotor performance in rocky habitats (Goodman, 2007; Kohlsdorf & Navas, 2007; Revell et al., 2007), I hypothesized that human alteration of the landscape should affect behavioral and morphological traits associated with locomotion. I tested for differences in jumping behavior, limb morphology, and sprinting performance between lizards living in areas with walls and areas without walls. The research demonstrates that human alteration of the environment can result in considerable microgeographic variation in important whole-organism traits such as locomotion.

Figure 1 The island of Naxos in the Greek Cyclades and representative pictures of the sites with and without walls.

I found significant differences in the body size (SVL) and hindlimb morphology of males (bold blue) and females (light red) from wall (top) and non-wall (bottom) sites. Total limb length was calculated from the sum of component parts, see Table 1 for segment-by-segment comparisons between the populations. Mean and standard error are presented for each measurement along with the p-value of the size-corrected LME model (see Table 1).

Methods

I collected 324 adult P. erhardii from 10, 50 m by 50 m sites within 15 km of each other on Naxos, a large island in the Greek Cyclade Islands. Five sites had stone walls, the other five were characterized by sandy substrate with interspersed Juniperus oxycedrus shrubs or a loose jumble of soil and Mediterranean phrygana (Fig. 1). All sites were selected for having a high density of lizards, and non-wall sites for being more than 200 m from the nearest built stone structure. While P. erhardii home range has not been investigated, sister species have reported home ranges no larger than 120 m2 (Brown, Gist & Taylor 1995; Swallow & Castilla, 1996) suggesting it is highly unlikely lizards from non-wall sites had originated on walls. For all lizards, I recorded sex and measured snout-to-vent length (SVL), and the length of each segment of the right fore and hind limb using digital calipers (Frankford Arsenal 672060).

I constructed two tracks for assessing lizard locomotion. Each track was 50 cm wide and 2 m long, with walls approximately 50 cm tall, constructed from heavy-duty plastic sheeting. One track had a sandy substrate (5 cm depth) reflecting the homefield of the five non-wall lizard populations, and the other was paved with large flagstones (averaging approximately 20 cm in diameter) from nearby walls. These flagstones were placed so each abutted the next, resulting in haphazard small (1–4 cm) gaps between uneven rock edges, mimicking the position and spacing of stones on top of local rock walls. The arrangement of rocks did not change through the entirety of the experiment. The track was not heated and conditions were kept consistent for all trials.

Before each trial, all lizards were allowed to thermoregulate at will for at least 30 min along a temperature gradient radiating from a suspended lamp (sand temperature 45 C–25 C). Immediately before running the lizard, I recorded their temperature using a cloacal thermometer (Miller and Webber T6000). The sprinting temperatures selected by males and females between wall and non-wall sites did not significantly differ (Males: wall: 29.7 ± 1.2 C, non-wall: 30.3 ± 1.9 C; Females: wall: 29.3 ± 1.2 C, non-wall: 29.2 ± 2.2 C). Lizards were stationary in the same start position at the beginning of each trial. Each sprint was recorded with a video camera (Sony HDRPJ260V; 1,920 × 1,080 px; 50 Hz) suspended directly over the track using a tripod. The camera’s field of view encompassed the first 1.5 m of track and had a full dorsal perspective of the running lizard. As motivation impacts sprint speed performance (Losos, Creer & Schulte, 2002; Irschick et al., 2005), if the lizard did not seemingly run maximally I discarded their trial during analysis.

I calculated the position of the lizard frame-by-frame relative to a tape measure in the field of view using a custom-built JavaScript program (code: https://github.com/bkazez/savra). To calculate velocity and acceleration, I fit a quintic spline to the position data (Walker, 1998) with the SPAPI function in MatLab (MathWorks Inc., 2014). This spline function was then differentiated such that the maximum of the first derivative yielded maximum velocity, the second derivative, maximum acceleration. Finally, I watched each stone-substrate trial and counted the number of times the lizards jumped (body and all limbs simultaneously in the air) from rock to rock. The Yale IACUC office approved all experiments involving animals (permit: 2013-11548). All work was conducted with permission from the Greek Ministry of Environment, Energy, and Climate Change (Permit 11665/1669).

Statistical analyses

To test for differences in morphology between populations I used linear mixed effects models, evaluated using the LME command within the NLME (v3.1-121; 2015) package in R (v3.1.2; 2014). As the morphometric and performance traits were not normally distributed, each was Log10 transformed before analysis. Each morphometric was treated as a response variable with presence or absence of wall as fixed effects and with site of origin as a random effect. I tested for relative morphological differences by adding SVL as a covariate. To test for differences in performance response variables—maximum velocity and acceleration over each substrate—I again used wall presence or absence as a fixed effect and site of origin as a random effect with sprint temperature as an additional random effect. Finally, to determine whether there was a difference in propensity to jump between the wall and non-wall populations, I used the count of jumps across the rocky experimental substrate as a fixed effect and included temperature as a random effect. A Shapiro–Wilk test determined that jump counts were not normally distributed (W = 0.9435, P < 0.0001), and so I Log10 transformed the jump counts for all analyses. Whenever body size or temperature was used in a model, they were standardized to have a mean of zero so as to make the estimates of each response variable directly interpretable (standardized value = initial value − global mean value). In all cases, males and females were analyzed independently to reduce interactions in the models. Finally, I used a type II ANOVA (CAR package, v2.0-25) to calculate Wald chi-square values for the model fixed effects and assign p-values appropriate for the unbalanced design (Langsrud, 2003). Figures were made in JMP (v11.2.0. SAS Institute Inc., 2013.).

Results

For clarity, all test statistics are related in the referenced tables. In-text, I instead present the average trait value, plus and minus the standard error. Lizards, both males and females, from wall sites had larger SVLs than lizards at non-wall sites (males: wall: 62.42 ± 0.62 mm, non-wall 58.13 ± 0.44 mm; females: wall: 59.23 ± 0.74 mm, non-wall: 55.02 ± 0.59 mm; Fig. 1; Table 1). This pattern was consistent across both sexes for multiple limb measurements (Fig. 1). In particular, the distal portions of the hindlimbs—the length between the ankle joint and the tip of the longest toe, and the longest toe itself—were relatively (standardized by SVL) longer among wall populations (Table 1). All together, lizards living on walls had proportionally longer hind limbs than lizards in non-wall habitats (Fig. 2A and Table 1).

Figure 2 Lizards from wall sites had proportionally longer hindlimbs, relative to SVL (A). These longer hindlimbs corresponded to significantly faster accelerations among females over a rocky experimental substrate (B), and to significantly increased jumping propensity for both males and females (C).

Shaded regions in (A) reflect 95% confidence intervals and standard error bars have been added for (B) and (C). All comparisons with (*) are significant p < 0.05.

I found no difference in maximum velocity among lizards from either habitat of origin across either experimental substrate (male maximum velocity on rock: wall: 1.99 ± 0.45 m/s, non-wall 1.98 ± 0.6 m/s; female maximum velocity on rock: wall: 1.83 ±0.44 m/s, non-wall 1.76 ± 0.31 m/s; male maximum velocity on sand: wall: 1.98 ± 0.6 m/s, non-wall: 1.82 ± 0.42 m/s; female maximum velocity on sand: wall: 1.73 ± 0.47 m/s, non-wall 1.67 ± 0.34 m/s; Table 2). While I found no difference in either population’s acceleration capacity over sand (see Table 2), I found that females from wall sites accelerated over the rocky experimental substrate faster than lizards from non-wall sites (Fig. 2B and Table 2).

Table 1 Results of the linear mixed effects models comparing morphological measurements between wall and non-wall lizard populations.

After SVL was shown to differ between sites, relative differences in morphology; that is, morphology standardized by SVL was tested. All morphometrics were Log10 transformed to meet assumptions of normality.

	Males	Females	
Morphometric:	N	X2	DF	p	N	X2	DF	p	
Model:	∼ Wall | Site	
SVL	175	9.017	1	0.0027*	149	4.343	1	0.0372*	
Model:	∼ Wall + SVL | Site	
Total arm length	175	0.010	1	0.9213	149	3.849	1	0.0498*	
Hip to knee	175	0.473	1	0.4918	149	1.018	1	0.3130	
Knee to ankle	175	0.066	1	0.7974	149	3.512	1	0.0609	
Ankle to tip of toe	175	5.226	1	0.0223*	149	9.016	1	0.0027*	
Longest toe	175	5.774	1	0.0163*	149	19.701	1	<0.0001*	
Total leg length	175	9.717	1	0.0018*	149	15.446	1	<0.0001*	
Notes.

* Denotes significance at the p < 0.05 level.

Table 2 Linear mixed effects models comparing performance and behavior between wall and non-wall lizard populations.

All performance and behavior metrics were Log10 transformed to meet assumptions of normality.

	Males	Females	
Performance metric:	N	X2	DF	p	N	X2	DF	p	
Model:	∼ Wall + SVL | Site + SprintTemp	
Max velocity rock	171	0.966	1	0.3256	143	1.577	1	0.2092	
Max acceleration rock	170	1.587	1	0.2078	143	7.024	1	0.0080*	
Max velocity sand	166	0.070	1	0.7915	142	0.389	1	0.5329	
Max acceleration sand	165	0.203	1	0.6526	141	0.128	1	0.7202	
Jumps	172	3.810	1	0.0481*	145	6.643	1	0.0099*	
Notes.

* Denotes significance at the p < 0.05 level.

In contrast to other studies, I found that individual SVL was not a significant predictor of maximum velocity or acceleration across either substrate for either sex (Table S1). Similarly, hindlimb length and the length of the longest toe did not significantly explain variation in sprinting performance (Table S1).

Both males and females from wall populations exhibited a strong behavioral shift: the lizards accustomed to walls consistently traversed the rocky experimental substrate by jumping rock-to-rock (see Video S1 and S2). Non-wall lizards jumped significantly fewer times crossing the same experimental track (males: wall: 3.4 ± 1.2 jumps, non-wall: 2.5 ± 1.2 jumps; females: wall: 3.4 ± 1.2 jumps, non-wall: 2.0 ± 1.1 jumps; Table 2 and Fig. 2C). Differences in jumping propensity were not explained by SVL (Table 3); however, jump counts were significantly informed by the length of the distal components of the hind limbs (male longest toe: p = 0.0181; female longest toe: p = 0.0108; Table 3).

Table 3 Results of regressions between three morphological variables and the count of jumps across the rocky substrate.

All variables were Log10 transformed to fit the assumption of normality.

	Jumps	
	Males	Females	
SVL	p = 0.2573	0.1557	
	R2Adj = 0.0017	0.0073	
Length of longest toe	p = 0.0181*	0.0108*	
	R2Adj = 0.027	0.0389	
Total hindlimb length	p = 0.1182	0.0031	
	R2Adj = 0.0085	0.0543	
Notes.

* Denotes significance at the p < 0.05 level.

Discussion

I found consistent differences between close-proximity populations of P. erhardii inhabiting different habitat-structure contexts. Lizards originating on sites with walls were larger than lizards from non-wall sites. Furthermore, the absolute length of each component of the hind limbs, and the relative length of the hindlimb as a whole was proportionally larger among wall populations of both sexes (Fig. 2A). The difference in relative hindlimb length was driven by proportional differences in the foot and longest toe of wall-inhabiting lizards (Table 1).

Morphological differences between lizard populations sometimes result in local, habitat-specific performance advantages (e.g., limb length determining motility across branches of different diameters in Anolis; Calsbeek & Irschick, 2007; Losos, 2011). Long limbs in Lacertids are in some species an adaptation for fast sprints over loose substrates (Bauwens et al., 1995; Bonine & Garland Jr, 1999). However, I found no inter-population differences in sprinting ability across sand. Alternatively, long hind limbs are also associated with jumping capacity and propensity (Moermond, 1979; Losos, 1990; Toro, Herrel & Irschick, 2004), particularly in rocky habitats (Goodman, 2007; Kohlsdorf & Navas, 2007; Revell et al., 2007). Indeed, I found that lizards from wall sites (with longest hind limbs) jump 1.5 times more often than non-wall populations on the same experimental track (Fig. 2C; Video S1 and S2). While there was no difference in the maximum sprint velocity of either population across either substrate, I found that females from rock wall populations accelerated more quickly than those from the non-wall habitats over the rocky experimental substrate (Fig. 2B).

Table 4 Average and standard deviation of the performance of lizards from wall and non-wall sites.

	Males	Females	
	Wall	No wall	Wall	No wall	
	Mean	±SD	Mean	±SD	Mean	±SD	Mean	±SD	
Velocity rock (m/s)	1.99	0.45	1.91	0.47	1.83	0.44	1.76	0.31	
Acceleration rock (m/s/s)	88.57	29.87	79.84	26.89	84.05	28.67	73.32	21.78	
Velocity sand (m/s)	1.98	0.60	1.82	0.42	1.73	0.47	1.67	0.34	
Acceleration sand (m/s/s)	87.88	38.68	81.00	30.60	77.82	29.49	77.73	35.55	

Motivation will considerably affect measurements of an animal’s performance in laboratory conditions (Losos, Creer & Schulte, 2002; Irschick et al., 2005). For that reason, many trials on relatively fewer individuals may provide clearer insights into maximal ability; however, even these results should be interpreted with caution (Losos, Creer & Schulte, 2002; Irschick et al., 2005). Due to logistical constraints, repeated trials were not possible for this study, and accordingly the high variation in performance obscured the trends predicted for the observed morphological differences between populations. Additionally, others have demonstrated that slow video frame rates are prone to errors in estimating acceleration of fast-moving animals (Walker, 1998). A 50 Hz camera was the maximum speed available for this field study, and, although my calculated values (Table 4) are commensurate with published values for closely related species (Vanhooydonck et al., 2015), further work with high-speed cameras (exceeding 250 Hz) and repeated trials will be necessary to show whether and how these observed morphological differences translate to performance differences.

Few studies investigate relative lizard locomotion capacity over multiple experimental substrates (Vanhooydonck et al., 2015). Studies that have did not find that species racing on an experimental substrate similar to their characteristic natural habitat necessarily performed better (Tulli, Abdala & Cruz, 2012; Vanhooydonck et al., 2015). This study suggests one potential explanation: the intraspecific differences observed here are commensurate with some published interspecific comparisons (Tulli, Abdala & Cruz, 2012; Vanhooydonck et al., 2015), meaning that variation among source populations could change the interpretation of interspecific comparisons. This study demonstrates that locomotor behavior and performance is contingent on the structure and ecology of source populations and not necessarily typological for a species.

Intraspecific context-dependence in locomotion morphometrics have been demonstrated between physically isolated populations (e.g., island vs mainland; Van Damme, Aerts & Vanhooydonck, 1998), and populations inhabiting dramatically different natural contexts (e.g., Des Roches et al., 2014). Coordinated intraspecific changes in locomotion behavior, morphology, and performance are strong indications that selection acts holistically on these traits across ecological contexts (Miles, Snell & Snell, 2001; Calsbeek & Irschick, 2007; Gifford, Herrel & Mahler, 2008). The differences related here in lizard morphology and jumping behavior over small spatial scales are noteworthy, and demonstrate the significant potential effect of anthropogenic microhabitat alteration on an important whole-organism trait, locomotion.

Supplemental Information

Video S1 Wall population sprint speed trial

Click here for additional data file.

Video S2 Non-wall population sprint speed trial

Click here for additional data file.

Supplemental Information 1 All data used for manuscript

Click here for additional data file.

Supplemental Information 2 R Code used for all analyses

Click here for additional data file.

Table S1 Results of correlations between three morphometrics and four performance traits

All morphometrics and performance measures were Log10 transformed to meet assumptions of normality.

Click here for additional data file.

Thanks go to P Pafilis and J Foufopoulos for logistical aid in-country; K Culhane, Z Miller, and A Mossman for help in the field; B Kazez and B Redding for video analysis assistance; and A Herrel, M Lambert, O Schmitz, and D Skelly for manuscript comments.

Additional Information and Declarations

Competing Interests

Author Contributions

Animal Ethics

Field Study Permissions

Data Availability

The author declares there is no competing interests.

Colin Donihue conceived and designed the experiments, performed the experiments, analyzed the data, contributed reagents/materials/analysis tools, wrote the paper, prepared figures and/or tables, reviewed drafts of the paper.

The following information was supplied relating to ethical approvals (i.e., approving body and any reference numbers):

All experiments involving animals were approved by Yale IACUC, 2013-11548.

The following information was supplied relating to field study approvals (i.e., approving body and any reference numbers):

All work was conducted with permission from the Greek Ministry of Environment, Energy, and Climate Change (Permit 11665/1669).

The following information was supplied regarding data availability:

The raw dataset has been included as Supplemental Information 1.

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
