# Peer review of "Microgeographic variation in locomotor traits among lizards in a human-built environment"

_PeerJ, doi:10.7717/peerj.1776_

## Round 0.1 · original submission · Major Revisions

Your manuscript is interesting and illustrates a trend in size among two groups of wall lizards. Both of the reviewers provided insight and constructive criticism. I also add my comments within the PDF that I have uploaded.

My comments are mostly about the biology of the animals. Particularly, I would like to better understand the possible causes and consequences of this different morphology. Is it facultative? That is, if you did a cross transplant experiment with young lizards, would the ones living on walls acquire the same morphology (and vice versa)? Or, is it simply social dominance and age that makes the difference?

I have several other comments within the text of the PDF, some of which were also suggested by the reviewers. I hope that the constructive ideas the two reviewers and I suggested help improve the quality of the manuscript.

·

Basic reporting

Line 18-19: The stated trend of distal components being proportionally longer is not clearly illustrated in results.
Line 20: “morphologies facilitated” There is a correlation here but no direct evidence of the trait facilitating the behavior. Having longer legs and number of times jumped are not necessarily linked. What is the biology being tested here?
Line 43: Intraspecific variation in performance across habitats has been studied in at least three other lizard systems. See the following literature as a starting point:
Intrapopulation variation in endurance of Galapagos lava lizards (Microlophus albemarlensis): evidence for an interaction between natural and sexual selection By: Miles, DB; Snell, HL; Snell, HM EVOLUTIONARY ECOLOGY RESEARCH 2001
The evolution of locomotor morphology, performance, and anti-predator behaviour among populations of Leiocephalus lizards from the Dominican Republic By:Gifford, ME; Herrel, A ; Mahler, DL
The quick and the dead: Correlational selection on morphology, performance, and habitat use in island lizards By:Calsbeek, R ; Irschick, DJ

Experimental design

The term population is used in a number of different contexts here. What is a population in this study, a sampling site, a habitat type and what is the biological justification of it?
Line 58-64: Do these lizards live on top of the walls or the sides of the walls, or in the cracks? More details of the habitat use in these different habitat classes would be very helpful for interpreting the choices made in the study and the outcomes. Was density similar in walled and wall-less sites? Do these lizards move 200m? Are the sampled sites different "populations?" What is the among "population" within habitat variation in traits?
Line 73: Should this be lamp?
Lines 77-85: What was the repeatability of individual performance? Were any runs repeated within individuals so this could be estimated? Were observations made on lizard’s apparent motivation? Was the substrate temperature controlled/ the same for each trial? For example, a very hot track could lead to more or less jumping? If a lizard didn’t jump from rock to rock how did it transverse the areas between the rocks? What size were the “gaps” between rocks? These need to be spelled out in the methods clearly.
Line 92: caps for model used later, need consistency.
Line 94: Did “site of origin” have any significant impacts? Is this different from how “population” is being used in for example line 129?
Line 112-118: What is important here are SVL adjusted differences, it is generally expected that larger lizards will have longer legs for within species samples.
Line 136: Which results are adjusted for SVL, only the one indicated? See comment about lines 112-118 above.
Results in general are largely a written out form of table 1. Is there a better way to summarize the key results here?
Lines 144-145: Where were proportional differences presented? Proportional to overall body size as in line 123? or to medial parts of the leg? Not clear here what is being indicated and I can’t really go into table 1 and easily see this.
Lines155-157: Was female reproductive condition controlled for in this population? Was this experiment done during the breeding season? Were any gravid? Were there potential hormone level differences? Performance drops dramatically in some populations following the reproductive season. This could drive some of the high variance seen if lizards were sampled across the season.
Lines 158-164: Given the high variance within groups could camera speed have been a really major problem for the methodology? There may be additional significant results hidden within this study if a faster camera had been used and runs were repeated for each individual.
Interesting and consistent trends are present but the significance is not. What may be driving the high within group variation, if anything?

Validity of the findings

Line 100-101: The normality test stat reported for jumping appears to indicate that the distribution is significantly different from normal. p<0.0001 rejects null hypothesis of normality...
The tests done here need to be more fully justified. For example, were the assumptions of using SVL as a covariate met? If temperature was not significantly different, what was the value of adding it as an additional effect? A number of the tests are not independent. Some tests include the same measurements as a subset. How many independent tests are actually being done here and is it setup in a way that does not impact the functional p-levels? This is not clear to me.
Lines 146-151: There is a relatively large literature that explores the relationships of morphology and performance that needs to be brought to bear on these results.
It is not clear to me what “opposite trend” means in this context given the broader literature on comparative studies of lizard limb length and sprint speed.
Lines 151-154: Not sure what the expected functional relationship between limb length and number of jumps should be. There could be an expected relationship between length of jump and limb length. What is the mechanism underlying this “prediction?” This needs to be developed in more depth.
Lines 166-168: "Studies that have, found little advantage in performance..." What is the advantage being referred to here?
Lines 242-245: (a) What are shaded regions of the graph? What method was used to generate this graph? (b) In legend - only for females of one type? This sounds like an overstatement of the results. Do the non-size corrected values need to be presented? What are the error bars?

Additional comments

This is a very interesting study system and you have a fair amount of data. You also have apparent trends that are interesting. By exploring what is driving the large within group variation you may be able to more fully uncover what is happening in this system.

·

Basic reporting

The manuscript is well written and adheres to the PeerJ formatting requirements.

Experimental design

The experimental design is generally strong, except for some areas where the statistical testing could be more explicit (see general comments below).

Validity of the findings

The presumed links between morphology and performance need to be explicitly tested (see general comments below).

Additional comments

Thank you for the opportunity to review this manuscript, “Microgeographic variability in locomotor traits among lizards in a human-built environment”. The manuscript describes the results of a well-done study investigating how “microgeographic” variation in the local environment of a single species (Aegean wall lizards) can affect limb morphology and locomotor performance. Particular strengths of the study include the impressive sampling effect (>300 individuals) and the use of naturalistic substrates, in lieu of standard laboratory proxies, to test locomotor performance. I expect this study to be well-cited once it is published. However, there are a few issues that should be addressed before the manuscript is ready for publication.

The only “major” issue I identified concerns the author’s repeated assertions that differences in limb morphology (where wall-living populations had absolutely and relatively longer hindlimbs) led to differences in performance (an increase in jumping proclivities and, among females, acceleration capacity). This statement is merely a hypothesis, and has not actually been tested in the context of the current research design. Whereas it was shown that wall-living lizards differed from more “terrestrial” (no-wall) populations in both morphology and performance, an explicit link between morphology and performance was never demonstrated. The morphological and performance differences could be coincidental, rather than causal. What is required here is an explicit statistical test of the possible linkages between limb length and jumping/acceleration potential. Why not include limb length metrics as fixed variables (covariates) in the LME performance models? Something like:

new.mod=lme(jump.count~wall.presence*limb.length…)

OR

new.mod=lme(jump.count~wall.presence*(limb.length+SVL)…)

if you want to control for the effects of overall body size.

If there are covariate*factor interactions in the model (i.e., if the relationship between the performance variable and the morphological variable varies between wall and no-wall populations), the it would be sufficient to test effects of limb length on performance independently within populations (i.e., within each population, do longer limb individuals show better performance):

wall.mod=lme(jump.count~limb.length…,data=subset(lizard.data,wall.presence==”wall”))

Or, you could even just test for linkages between limb length and performance across the whole sample, ignoring habitat. Regardless, the presumed morphology-performance gradient should be explicitly addressed.

MINOR ISSUES (line numbers in parentheses)

(32) change “and individual’s” to “an individual’s”

(33) insert a comma between “habitat” and “substrate”

(61-62) The author states that “wall-less sites were at least 200m from the nearest built-stone structure”. Any idea what the home range size of this species is? Would they never range far enough to get to stone structures?

(64, and Fig. 1) Fig 1 seems to indicate that total limb lengths were measured curvilinearly. How was this accomplished with calipers? Or was total limb length just calculated as the sum of all individual segments?

(73) As much fun as I had picturing lizards thermoregulating under a suspended “lamb”, I believe the author meant to say “lamp” here.

(94) Why not include individual nested within site as another random factor? I think the syntax would be:

new.mod=lme(jump.count~wall.presence*limb.length, random=1|site/individual)

(106-107) Why did you choose not to test for gender-based interactions in the model? My default would be to test for such interactions, and then split into separate male/female models where appropriate after that. Gestalt models should be preferred over piecemeal models (in my opinion).

(Results, in general) There’s no need to separately list chi-squared values, p-values, etc. in the text where you have all the same data in Table 1. For the sake of readability, I recommend removing all of the statistical reporting from the text and have a blanket statement at the beginning of the results saying something like, “Values for all test statistics, degrees of freedom, sample sizes, and p-values are reported in Table 1”. In the Results text, then go on to say what these statistical tests indicate.

(158-164) The author is certainly correct to point out the possible error inherent in using a relatively low frame rate. Was it not possible to split the frames into fields, effectively doubling the frame rate to 100fps? JES Deinterlacer is freeware program that could do this, for instance (http://jeschot.home.xs4all.nl). Nevertheless, if the author digitized the locomotor sequence over the entire 1.5m visible in the camera field of view, I would expect the aliasing errors due to using a low fps would mostly be attenuated. I wouldn’t want to estimate velocity, and especially acceleration, over a relatively short interval with such a low fps. If, however, the reported velocity and acceleration values are averages across a longer sequence of movement, aliasing shouldn’t be much of an issue.

(Fig. 1, Table 1 and elsewhere) “Arm” and “Leg” are specific anatomical terms (i.e., brachium and crus). When referring to the entire limb, I recommend using “forelimb” and “hindlimb”.

---

## Round 0.2 · Minor Revisions

Both reviewers and I agree that the quality of the text has improved. A few minor details remain, such as the use of standard deviations, standard errors, and in my humble opinion, the appropriate confidence interval. The choice of which one to use should be consistent and relevant. The standard deviation is about the variability of the variable itself. The standard error is about the variability of the average value of the populations. The confidence interval allows us to compare populations, which seems to be the idea you wish to convey. In your methods, you say you use the standard deviation, while in the figures, you use the standard error. I'd only suggest that you make those choices very clear.

·

Basic reporting

Revision is much more clear. Size is reported as mean and standard deviation in text but same numbers are given as mean and SE in figure 1.

Experimental design

No comment

Validity of the findings

Focus is now on key results and much more accessible to the reader.

Additional comments

No comment.

·

Basic reporting

This manuscript was well-written in its first draft, and has only been improved upon revision.

Experimental design

The experimental design is sound, and any areas of weakness (i.e., the low frame rate of the camera used to record locomotion) have been adequately addressed. The author has done much to ameliorate my previous concerns about the analytical design.

Validity of the findings

Overall, the findings appear valid. As I commented in my first review, I expect this study to be well-cited in the ecological performance literature.

---

## Round 0.3 · accepted · Accept

Thank you for your new and improved version.